

# Relationship between perceptual and mechanical markers of fatigue during bench press and bench pull exercises: impact of inter-set rest period length

Danica Janicijevic[1,2], Sergio Miras-Moreno[3],
Maria Dolores Morenas-Aguilar[3], Pablo Jiménez-Martínez[4,5],
Carlos Alix-Fages[4,5,6] and Amador García-Ramos[3,7]

[1] Faculty of Sports Science, Ningbo University, Ningbo, China
[2] Research Academy of Human Biomechanics, The Affiliated Hospital of Medical School of Ningbo University, Ningbo, China
[3] Department of Physical Education and Sport, Universidad de Granada, Granada, España
[4] Research Group in Prevention and Health in Exercise and Sport (PHES), University of Valencia, Valencia, Spain
[5] ICEN Institute, Madrid, Spain
[6] Applied Biomechanics and Sport Technology Research Group, Autonomous University of Madrid, Madrid, Spain
[7] Department of Sports Sciences and Physical Conditioning, Faculty of Education, Universidad Catolica de la Santísima Concepcion, Concepcion, Chile

Corresponding author
Amador García-Ramos,
amagr@ugr.es

## ABSTRACT

This study aimed to explore whether the relationship between perceptual (rating of perceived exertion; RPE) and mechanical (maximal number of repetitions completed [MNR], fastest set velocity, and mean velocity decline) variables is affected by the length of inter-set rest periods during resistance training sets not leading to failure. Twenty-three physically active individuals (15 men and eight women) randomly completed 12 testing sessions resulting from the combination of two exercises (bench press and bench pull), three inter-set rest protocols (1, 3, and 5 min), and two minimal velocity thresholds (farther from muscular failure [$MVT_{0.45}$ for bench press and $MVT_{0.65}$ for bench pull] and closer to muscular failure [$MVT_{0.35}$ for bench press and $MVT_{0.55}$ for bench pull]). The duration of inter-set rest periods did not have a significant impact on RPE values ($p$ ranged from 0.061 to 0.951). Higher proximities to failure, indicated by lower MVTs, were associated with increased RPE values ($p < 0.05$ in 19 out of 24 comparisons). Moreover, as the number of sets increased, an upward trend in RPE values was observed ($p < 0.05$ in seven out of 12 comparisons). Finally, while acknowledging some inconsistencies, it was generally observed that higher magnitudes of the mechanical variables, especially MNR ($r_s < -0.55$ in three out of four comparisons), were associated with lower RPE values. These results, which were comparable for the bench press and bench pull exercises, suggest that post-set RPE values are affected by the fatigue experienced at both the beginning and end of the set.

## INTRODUCTION

During physical exercise, such as lifting weights, humans experience certain magnitudes of exertion (*Robertson et al., 2003*). While the terms "effort" and "exertion" are often mistakenly used interchangeably, they have different meanings. "Effort" refers to the quantifiable amount of mental or physical energy dedicated to a task, whereas "exertion" pertains to the subjective perception of the intensity and strenuousness experienced during a physical activity (*Abbiss et al., 2015*; *Halperin & Emanuel, 2020*). In this sense, effort is more aligned with the corollary discharge perspective while exertion is more aligned with the afferent feedback (*Halperin & Emanuel, 2020*). Rating of perceived exertion (RPE) scales have been widely used in sport-related research to subjectively evaluate the level of stress caused by various physical tasks (*Pageaux, 2016*; *Halperin & Emanuel, 2020*). While RPE has been predominantly utilized in the context of aerobic exercise, this metric has also been systematically applied to monitor resistance training sessions for several decades (*Robertson et al., 2003*; *Mayo, Iglesias-Soler & Kingsley, 2019*). In this regard, RPE has been recommended for monitoring perceptual responses associated with manipulating various acute resistance training variables such as load (*Hollander et al., 2008*) or the arrangement of rest periods (traditional, cluster, or inter-repetition rest designs) during sessions where the intensity of the load and the work-to-rest ratio remain constant (*Mayo, Iglesias-Soler & Kingsley, 2019*).

Reductions in muscle force production capacity can be attributed to processes occurring at or distal to the neuromuscular junction (*i.e.*, peripheral fatigue) or within the central nervous system itself (*i.e.*, central fatigue) (*Alix-Fages et al., 2022*). A recent study showed that the administration of opioids such as fentanyl (afferent blockage) attenuates group III and IV afferent feedback, resulting in improved physical performance and reduced RPE values during an intermittent knee-extensor all-out exercise (*Broxterman et al., 2018*). The increased firing of group III and IV muscle afferents, caused by mechanical forces and metabolite accumulation, has been well-documented to contribute to the development of central fatigue, affecting both spinal and supraspinal levels (*Amann et al., 2011*; *Taylor et al., 2016*; *Blain et al., 2016*; *Sidhu et al., 2017*). The level of metabolite accumulation can vary depending on how the acute resistance training variables are manipulated (*Alix-Fages et al., 2022*). For example, shorter inter-set rest periods (*Jukic et al., 2020*) and stopping sets closer to failure (*Sánchez-Medina & González-Badillo, 2011*) contribute to increasing metabolic stress, elevated RPE values, and greater mechanical fatigue during resistance training. Given the contributions of both afferent feedback and brain circuitry to the development of RPE values (*Pageaux, 2016*), and their susceptibility to the programming of resistance training variables (*Jukic et al., 2020*), it is both reasonable and important to investigate the effects of manipulating various acute resistance training variables on RPE.

The OMNI-Resistance Exercise Scale (OMNI-RES) of RPE has been shown to be sensitive to changes in multiple resistance training variables (*Robertson et al., 2003*). A linear relationship has been reported between loading intensities (30%, 60%, and 90% of the 1-repetition maximum (1RM)) and both RPE and electromyography activity during a leg extension exercise (*Duncan, Al-Nakeeb & Scurr, 2006*). Proximity to muscular failure

also influenced the development of RPE, with maximal set configurations resulting in higher RPE values (9–10) compared to submaximal set configurations (*Hiscock et al., 2016*). However, when sets are performed to volitional failure, RPE is likely to remain consistent regardless of the loading intensity and the inter-set rest duration (*Hiscock et al., 2016*). In this regard, *Mayo, Iglesias-Soler & Kingsley (2019)* explored the effect of three different set configurations equated by loading intensity (10RM), number of repetitions (40) and total rest (12 min) on RPE values: (i) inter-repetition rest configuration (40 individual repetitions separated by 18.5 s of rest), (ii) cluster configuration (10 sets of four repetitions with 80 s of inter-set rest), and (iii) traditional configuration (five sets of eight repetitions with 180 s of inter-set rest). Their main findings revealed that the inclusion of shorter but more frequent rest periods (*i.e.*, inter-repetition rest design) promoted lower RPE values compared to the cluster and traditional configurations, whereas a negative correlation was generally observed between RPE and the mean velocity of the training session. However, to the best our knowledge, previous research has not explored whether RPE values are also influenced by the duration of inter-set rest periods during resistance training sessions that do not involve failure, where sets are terminated upon reaching different submaximal velocity thresholds. Furthermore, it remains unknown whether the length of inter-set rest periods can impact the relationship between perceptual (RPE) and mechanical fatigue markers.

To address these research gaps, the main objective of the present study was to elucidate whether RPE is affected by the length of the inter-set rest periods during multiple sets of the bench press and bench pull exercises with cessation points at different proximities to muscular failure. The secondary objective was to explore the association between RPE and several mechanical variables such as the maximum number of repetitions completed (MNR), fastest mean velocity of the set ($MV_{fastest}$), and mean velocity decline (MVD). Our main hypothesis was that, regardless of the exercise and proximity to failure, RPE values would not be affected by the length of inter-set rest periods. We also hypothesised that RPE would be negatively correlated with MNR, $MV_{fastest}$, and MVD because all three mechanical variables are expected to be lower when subjects begin the sets more fatigued.

# MATERIALS AND METHODS

## Study design

A crossover study design was used to explore whether the relationship between perceptual (RPE) and mechanical (MNR, $MV_{fastest}$, and MVD) markers of fatigue is affected by the length of inter-set rest periods during multiple sets of the bench press and bench pull exercises performed until exceeding different absolute MVTs. Participants completed 12 testing sessions held twice weekly, with each session spaced at least 48 h apart but not more than 120 h to ensure adequate recovery. The 12 testing sessions resulted from the combination of two exercises (bench press and bench pull), three inter-set rest protocols (1 (R1), 3 (R3), and 5 (R5) min of rest), and two proximities to failure (farther from muscular failure ($MVT_{0.45}$ for bench press and $MVT_{0.65}$ for bench pull) and closer to muscular failure ($MVT_{0.35}$ for bench press and $MVT_{0.55}$ for bench pull)). The order of the 12 testing sessions was randomised using the Research Randomizer website

(https://www.randomizer.org). The testing sessions were conducted in the university research laboratory, and careful consideration was given to schedule each participant's sessions at the same time of day to minimize the potential influence of diurnal variations on strength performance (*Knaier et al., 2019*). Prior to each session, participants were queried regarding their readiness to engage in maximal resistance training. If a participant indicated that they did not feel prepared to perform at their maximal capacity for any reason (pain derived from the menstrual cycle, inadequate sleep, residual fatigue from other activities, *etc.*), the session was rescheduled. This approach allowed us to maintain the integrity of the study's focus on maximal performance, while also respecting the individual's physical state at the time of the session.

## Subjects

A total of 23 physically active individuals, comprising 15 men (age: 23.7 ± 4.3 years; body mass: 79.7 ± 10.7 kg; body height: 1.79 ± 0.08 m; bench press relative to body mass 1RM: 1.11 ± 0.23 kg·kg$^{-1}$; bench pull relative to body mass 1RM: 1.08 ± 0.15 kg·kg$^{-1}$) and eight women (age: 25.9 ± 8.7 years; body mass: 61.6 ± 6.0 kg; body height: 1.64 ± 0.04 m; bench press relative to body mass 1RM: 0.67 ± 0.10 kg·kg$^{-1}$; bench pull relative to body mass 1RM: 0.81 ± 0.10 kg·kg$^{-1}$), volunteered to participate in this study. Our participants were not only regularly engaged in physical activities but also had a background in resistance training, both independently and through prior projects conducted by our research team. We carefully selected those who demonstrated proficiency in the bench press and bench pull exercises, ensuring they could execute these exercises at maximum intended velocity with correct technique. Given their previous participation in our research group's activities, where they became well-acquainted with both the exercises in question and the use of RPE scales, we deemed an additional familiarization session for the current study to be redundant. None of the participants reported any physical limitations that could compromise the study results. Participants were explicitly instructed to abstain from any vigorous physical activity for 48 h prior to each laboratory visit and to avoid the intake of stimulant beverages, such as those containing caffeine, for at least 12 h before each testing session. Prior to participation, all participants were informed about the purpose and procedures of the study, and signed the informed consent form. The study protocol adhered to the principles outlined in the Declaration of Helsinki and was approved by the Institutional Review Board of the University of Granada (2046/CEIH/2021).

## Procedures

At the onset of each session, a general warm-up routine was implemented, consisting of jogging and dynamic stretching exercises. Once the warm-up was completed, an incremental loading test was conducted for the targeted exercise (bench press or bench pull) following standard procedures (*García-Ramos et al., 2019*; *Janicijevic et al., 2021*). Initially, a load of 20 kg was set for both exercises. Subsequently, the load was progressively increased in increments of 10 kg until the MV dropped below 0.50 m·s$^{-1}$ in the bench press or 0.80 m·s$^{-1}$ in the bench pull. Once this MV threshold was reached, further load adjustments were made in steps of 5 to 1 kg until the 1RM was successfully achieved.

All 1RMs were obtained at a MV $\leq 0.23$ m·s$^{-1}$ for the bench press and MV $\leq 0.56$ m·s$^{-1}$ for the bench pull. Participants executed two repetitions with light to moderate loads (MV $\geq$ 0.50 m·s$^{-1}$ for bench press and MV $\geq 0.80$ m·s$^{-1}$ for bench pull) and one repetition was performed with heavier loads (MV $< 0.50$ m·s$^{-1}$ for bench press and MV $< 0.80$ m·s$^{-1}$ for bench pull). To allow for adequate recovery, a rest period of 3 min was provided for light to moderate loads, while a rest period of 5 min was allocated for heavier loads.

Following the determination of their 1RM, subjects were given a rest period of 10 min before commencing the first set of the training session. The 1RM was directly assessed during each of the 12 experimental sessions. A load equivalent to 75% of the previously determined 1RM was used in the four subsequent sets of the testing session. Note that the 12 testing sessions varied in the exercise (bench press or bench pull), inter-set rest protocol (1 (R1), 3 (R3), and 5 (R5 min of rest)) or proximity to failure (farther from muscular failure (MVT$_{0.45}$ for bench press and MVT$_{0.65}$ for bench pull) or closer to muscular failure (MVT$_{0.35}$ for bench press and MVT$_{0.55}$ for bench pull)). Subjects were instructed to perform the concentric phase of all repetitions at maximal intended velocity and to terminate the set once a repetition was executed at a MV lower than the specific MVT established for that particular session. The terminal MVT in the bench press was $0.41 \pm 0.04$ m·s$^{-1}$ for the MVT$_{0.45}$ condition and $0.31 \pm 0.04$ m·s$^{-1}$ for the MVT$_{0.35}$ condition, whereas in the bench pull was $0.62 \pm 0.02$ m·s$^{-1}$ for the MVT$_{0.65}$ condition and $0.54 \pm 0.04$ m·s$^{-1}$ for the MVT$_{0.55}$ condition. The MVTs were chosen to ensure a similar reduction in MV for both the bench press and bench pull exercises, with reductions approximately at 0.12 and 0.22 ms$^{-1}$ for the sets terminated farther and closer to muscular failure, respectively. The terminal MVTs were higher for the bench pull compared to the bench press, reflecting the inherently greater MV values associated with both the 75% 1RM and the 1RM trials in the former exercise (*González-Badillo & Sánchez-Medina, 2010*; *García-Ramos et al., 2019*). Real-time MV feedback was provided immediately after each repetition to increase motivation, competitiveness, and mechanical performance (*Weakley et al., 2021*).

## Measurement equipment and data analysis

The bench press and bench pull exercises were performed in a Smith machine (Multipower Fitness Line, Peroga, Murcia, Spain). During the bench press exercise, subjects adhered to the standardized 5-point body position and touch-and-go technique. During the bench pull exercise, the barbell was deliberately paused for 1–2 s on the telescopic holders of the Smith machine when both elbows were fully extended and immediately after subjects were instructed to pull the barbell until it made contact with the bottom surface of the bench (thickness of 11.0 cm).

A validated linear position transducer (GymAware RS, Kinetic Performance Technologies, Canberra, Australia) was vertically mounted to the Smith machine's barbell and provided the MV of each repetition (*Weakley et al., 2021*). The following mechanical variables were considered in the present study: (i) the maximum number of repetitions completed before exceeding different MVTs (MNR), (ii) fastest MV of the set (MV$_{fastest}$), and (iii) mean velocity decline (MVD [%] = [MV$_{last}$ − MV$_{fastest}$] / MV$_{fastest}$ × 100). MV$_{last}$

represent the MV of the last repetition of the set. The MNR, $MV_{fastest}$, and $MV_{fast}$ for each set and exercise is provided in Table 1. Finally, the OMNI-Resistance Exercise Scale (OMNI-RES) of perceived exertion was used as a measure of perceptual fatigue (*Robertson et al., 2003*). A printed pictograph of the OMNI-RES scale was shown to the subjects 15 s after completing each set to report their RPE value (0–10), where 0 is extremely easy and 10 represents extremely hard.

## Statistical analyses

Descriptive data are presented as means and standard deviations. Given the departure from normal distribution for RPE values detected by the Shapiro-Wilk test ($p < 0.05$), non-parametric tests were utilized in this study. More specifically, the Wilcoxon signed-rank test was used to compare RPE values between the absolute MVTs (bench press: $MVT_{0.45}$ *vs.* $MVT_{0.35}$; bench pull: $MVT_{0.65}$ *vs.* $MVT_{0.55}$) separately for each exercise, inter-set rest protocol, and set number. The Friedman test with Bonferroni *post hoc* corrections was used to compare RPE values between (i) the inter-set rest protocols (R1 *vs.* R3 *vs.* R5) separately for each exercise, proximity to failure, and set number, and (ii) the set number (set 1 *vs.* set 2 *vs.* set 3 *vs.* set 4) separately for each exercise, inter-set rest protocol, and proximity to failure. In addition, the Spearman's correlation coefficient ($r_s$) was used to explore the association between perceptual (RPE) and mechanical (MNR, $MV_{fastest}$, and MVD) markers of fatigue obtained separately for each exercise and proximity to failure. It is important to note that perceptual and mechanical markers of fatigue were assessed while participants utilized varying inter-set rest periods. The scale used to interpret the magnitude of the $r_s$ was: trivial (0.00–0.09), small (0.10–0.29), moderate (0.30–0.49), large (0.50–0.69), very large (0.70–0.89), nearly perfect (0.90–0.99), and perfect (1.00) (*Hopkins et al., 2009*). All statistical analyses were performed using SPSS software version 25.0 (SPSS Inc., Chicago, IL, USA) and statistical significance was set at an alpha level of 0.05.

## RESULTS

Table 2 depicts the results of the Friedman and Wilcoxon tests comparing RPE values between the sets, MVTs, and inter-set rest protocols, while the pairwise comparisons are presented in Fig. 1. Significant differences among the sets were detected in seven out of 12 comparisons due to a consistent upward trend in RPE values as the number of sets increased. The effect of set was more recurrent in the bench pull (five out of six comparisons) than in the bench press (two out of six comparisons), but it was not consistently affected by the inter-set rest protocol or MVT. Significant differences between the MVTs were observed in 19 out of 24 comparisons due to consistent greater RPE values with the use of closer proximities to failure. The bench press exercise using the R1 protocol demonstrated the least disparity between the two proximities to failure, with significant differences observed solely in the fourth set. No significant differences in RPE values between the three inter-set rest protocols were observed for any exercise regardless of the set number or proximity to failure.

The relationships between RPE and mechanical variables (MNR, $MV_{fastest}$, and MVD) are depicted in Fig. 2. Large and negative correlations were generally observed between

**Table 1 Descriptive values (mean ± standard deviation) of the maximum number of repetitions completed before exceeding different minimal velocity thresholds (MNR), fastest mean velocity (MV) of the set ($MV_{fastest}$), and MV of the last repetition.**

| Variable | Exercise | Inter-set rest | MVT | Set number | | | |
|---|---|---|---|---|---|---|---|
| | | | | Set 1 | Set 2 | Set 3 | Set 4 |
| MNR | Bench press | R1 | $MVT_{0.45}$ | 4.0 ± 1.8 | 3.0 ± 1.4 | 2.8 ± 1.0 | 2.9 ± 1.2 |
| | | | $MVT_{0.35}$ | 6.1 ± 2.3 | 4.0 ± 1.5 | 3.2 ± 1.1 | 3.4 ± 0.7 |
| | | R3 | $MVT_{0.45}$ | 3.4 ± 2.0 | 3.3 ± 1.3 | 3.0 ± 1.8 | 3.4 ± 1.4 |
| | | | $MVT_{0.35}$ | 6.1 ± 1.6 | 5.2 ± 1.5 | 4.7 ± 1.3 | 4.6 ± 1.1 |
| | | R5 | $MVT_{0.45}$ | 3.8 ± 2.0 | 3.8 ± 1.5 | 4.0 ± 1.8 | 3.7 ± 1.4 |
| | | | $MVT_{0.35}$ | 5.9 ± 1.9 | 5.7 ± 1.6 | 5.3 ± 1.6 | 5.1 ± 2.1 |
| | Bench pull | R1 | $MVT_{0.65}$ | 7.6 ± 2.8 | 5.9 ± 2.3 | 5.6 ± 2.0 | 5.1 ± 1.5 |
| | | | $MVT_{0.55}$ | 10.2 ± 2.9 | 6.7 ± 1.4 | 6.0 ± 1.8 | 5.7 ± 1.7 |
| | | R3 | $MVT_{0.65}$ | 6.7 ± 3.0 | 6.3 ± 3.2 | 6.2 ± 2.5 | 6.2 ± 2.7 |
| | | | $MVT_{0.55}$ | 10.0 ± 3.2 | 9.6 ± 2.7 | 8.6 ± 2.3 | 8.2 ± 2.5 |
| | | R5 | $MVT_{0.65}$ | 7.2 ± 2.9 | 7.3 ± 2.7 | 7.0 ± 2.7 | 6.8 ± 2.4 |
| | | | $MVT_{0.55}$ | 10.9 ± 3.1 | 10.7 ± 3.6 | 9.9 ± 2.5 | 9.7 ± 3.7 |
| $MV_{fastest}$ ($m \cdot s^{-1}$) | Bench press | R1 | $MVT_{0.45}$ | 0.53 ± 0.06 | 0.50 ± 0.05 | 0.50 ± 0.05 | 0.49 ± 0.05 |
| | | | $MVT_{0.35}$ | 0.52 ± 0.08 | 0.45 ± 0.06 | 0.44 ± 0.06 | 0.45 ± 0.04 |
| | | R3 | $MVT_{0.45}$ | 0.51 ± 0.07 | 0.51 ± 0.05 | 0.50 ± 0.07 | 0.51 ± 0.06 |
| | | | $MVT_{0.35}$ | 0.52 ± 0.06 | 0.50 ± 0.06 | 0.49 ± 0.05 | 0.48 ± 0.05 |
| | | R5 | $MVT_{0.45}$ | 0.52 ± 0.07 | 0.52 ± 0.05 | 0.53 ± 0.06 | 0.51 ± 0.06 |
| | | | $MVT_{0.35}$ | 0.52 ± 0.06 | 0.51 ± 0.06 | 0.50 ± 0.06 | 0.50 ± 0.06 |
| | Bench pull | R1 | $MVT_{0.65}$ | 0.78 ± 0.05 | 0.76 ± 0.04 | 0.75 ± 0.04 | 0.75 ± 0.04 |
| | | | $MVT_{0.55}$ | 0.76 ± 0.06 | 0.72 ± 0.05 | 0.70 ± 0.04 | 0.71 ± 0.04 |
| | | R3 | $MVT_{0.65}$ | 0.76 ± 0.05 | 0.76 ± 0.06 | 0.76 ± 0.05 | 0.76 ± 0.06 |
| | | | $MVT_{0.55}$ | 0.76 ± 0.05 | 0.77 ± 0.05 | 0.77 ± 0.05 | 0.75 ± 0.06 |
| | | R5 | $MVT_{0.65}$ | 0.77 ± 0.06 | 0.77 ± 0.06 | 0.77 ± 0.06 | 0.77 ± 0.05 |
| | | | $MVT_{0.55}$ | 0.75 ± 0.05 | 0.78 ± 0.06 | 0.77 ± 0.06 | 0.76 ± 0.07 |
| $MV_{last}$ ($m \cdot s^{-1}$) | Bench press | R1 | $MVT_{0.45}$ | 0.41 ± 0.04 | 0.40 ± 0.04 | 0.40 ± 0.04 | 0.40 ± 0.03 |
| | | | $MVT_{0.35}$ | 0.30 ± 0.04 | 0.32 ± 0.04 | 0.32 ± 0.02 | 0.28 ± 0.05 |
| | | R3 | $MVT_{0.45}$ | 0.41 ± 0.03 | 0.41 ± 0.03 | 0.41 ± 0.03 | 0.40 ± 0.05 |
| | | | $MVT_{0.35}$ | 0.30 ± 0.03 | 0.31 ± 0.04 | 0.31 ± 0.03 | 0.30 ± 0.04 |
| | | R5 | $MVT_{0.45}$ | 0.41 ± 0.03 | 0.42 ± 0.03 | 0.41 ± 0.04 | 0.40 ± 0.05 |
| | | | $MVT_{0.35}$ | 0.31 ± 0.03 | 0.30 ± 0.03 | 0.30 ± 0.03 | 0.30 ± 0.04 |
| | Bench pull | R1 | $MVT_{0.65}$ | 0.63 ± 0.02 | 0.62 ± 0.02 | 0.62 ± 0.03 | 0.63 ± 0.03 |
| | | | $MVT_{0.55}$ | 0.53 ± 0.04 | 0.54 ± 0.04 | 0.53 ± 0.06 | 0.56 ± 0.04 |
| | | R3 | $MVT_{0.65}$ | 0.63 ± 0.02 | 0.62 ± 0.02 | 0.62 ± 0.02 | 0.62 ± 0.02 |
| | | | $MVT_{0.55}$ | 0.53 ± 0.04 | 0.54 ± 0.02 | 0.54 ± 0.04 | 0.55 ± 0.04 |
| | | R5 | $MVT_{0.65}$ | 0.62 ± 0.02 | 0.63 ± 0.02 | 0.62 ± 0.01 | 0.62 ± 0.02 |
| | | | $MVT_{0.55}$ | 0.53 ± 0.06 | 0.53 ± 0.02 | 0.54 ± 0.03 | 0.53 ± 0.04 |

**Note:**
  MVT, minimum velocity threshold; R1, 1 min of inter-set rest; R3, 3 min of inter-set rest; R5, 5 min of inter-set rest.

**Table 2  Ratings of perceived exertion (RPE) values reported during resistance training sessions conducted with the bench press and bench pull exercises.**

| Exercise | Inter-set rest | MVT | Set 1 | Set 2 | Set 3 | Set 4 | Set | MVT Set 1 | MVT Set 2 | MVT Set 3 | MVT Set 4 | Rest Set 1 | Rest Set 2 | Rest Set 3 | Rest Set 4 |
|---|---|---|---|---|---|---|---|---|---|---|---|---|---|---|---|
| Bench press | R1 | $MVT_{0.45}$ | 6.2 ± 1.2 | 6.3 ± 1.0 | 6.2 ± 1.4 | 6.3 ± 1.7 | 0.436 | 0.202 | 0.137 | 0.165 | **0.046** | 0.260 | 0.846 | 0.685 | 0.433 |
| | | $MVT_{0.35}$ | 6.5 ± 1.5 | 6.8 ± 1.4 | 6.9 ± 1.7 | 7.0 ± 1.5 | **0.018** | | | | | 0.061 | 0.661 | 0.681 | 0.729 |
| | R3 | $MVT_{0.45}$ | 5.7 ± 1.9 | 6.0 ± 1.6 | 5.7 ± 2.0 | 6.0 ± 1.9 | 0.183 | **0.006** | **0.017** | **0.016** | **0.004** | | | | |
| | | $MVT_{0.35}$ | 6.6 ± 1.2 | 6.9 ± 1.2 | 6.7 ± 1.3 | 6.8 ± 1.5 | 0.344 | | | | | | | | |
| | R5 | $MVT_{0.45}$ | 5.6 ± 2.0 | 5.7 ± 1.9 | 5.8 ± 2.0 | 5.9 ± 2.0 | 0.213 | 0.081 | **0.007** | **0.006** | 0.070 | | | | |
| | | $MVT_{0.35}$ | 6.2 ± 1.6 | 6.7 ± 1.1 | 6.8 ± 1.1 | 6.7 ± 1.1 | **0.032** | | | | | | | | |
| Bench pull | R1 | $MVT_{0.65}$ | 6.3 ± 1.3 | 6.4 ± 1.3 | 6.6 ± 1.5 | 6.8 ± 1.6 | **0.003** | **0.022** | **0.002** | **0.027** | **0.009** | 0.394 | 0.945 | 0.776 | 0.424 |
| | | $MVT_{0.55}$ | 7.2 ± 1.3 | 7.5 ± 1.1 | 7.5 ± 1.3 | 7.6 ± 1.4 | 0.060 | | | | | 0.951 | 0.338 | 0.382 | 0.554 |
| | R3 | $MVT_{0.65}$ | 5.9 ± 1.6 | 6.0 ± 1.9 | 6.4 ± 1.7 | 6.4 ± 2.0 | **0.011** | **<0.001** | **<0.001** | **0.001** | **0.001** | | | | |
| | | $MVT_{0.55}$ | 7.3 ± 1.1 | 7.7 ± 0.9 | 7.8 ± 1.1 | 7.7 ± 1.3 | **0.004** | | | | | | | | |
| | R5 | $MVT_{0.65}$ | 5.9 ± 1.4 | 6.2 ± 1.2 | 6.5 ± 1.5 | 6.5 ± 1.5 | **0.004** | **<0.001** | **<0.001** | **0.004** | **0.022** | | | | |
| | | $MVT_{0.55}$ | 7.1 ± 1.1 | 7.4 ± 1.2 | 7.3 ± 1.2 | 7.3 ± 1.7 | **0.038** | | | | | | | | |

**Note:**
Mean ± standard deviation. MVT, minimum velocity threshold; R1, 1 min of inter-set rest; R3, 3 min of inter-set rest; R5, 5 min of inter-set rest. Bold numbers indicate significant differences ($p < 0.05$).

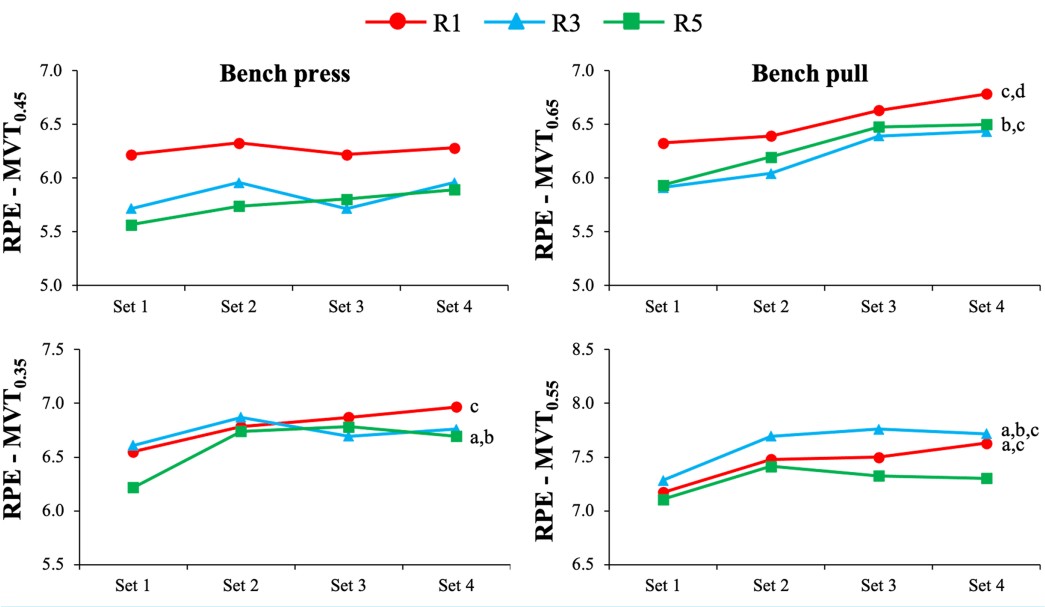

**Figure 1  Comparison of RPE between the different inter-set rest periods and sets during the bench press (left) and bench pull (right) exercises performed with moderate (upper) and high (lower) levels of effort.** MVT, minimum velocity threshold; R1, 1 min of inter-set rest; R3, 3 min of inter-set rest; R5, 5 min of inter-set rest; a, significant differences between set 1 and set 2; b, significant differences between set 1 and set 3; c, significant differences between set 1 and set 4; d, significant differences between set 2 and set 4.

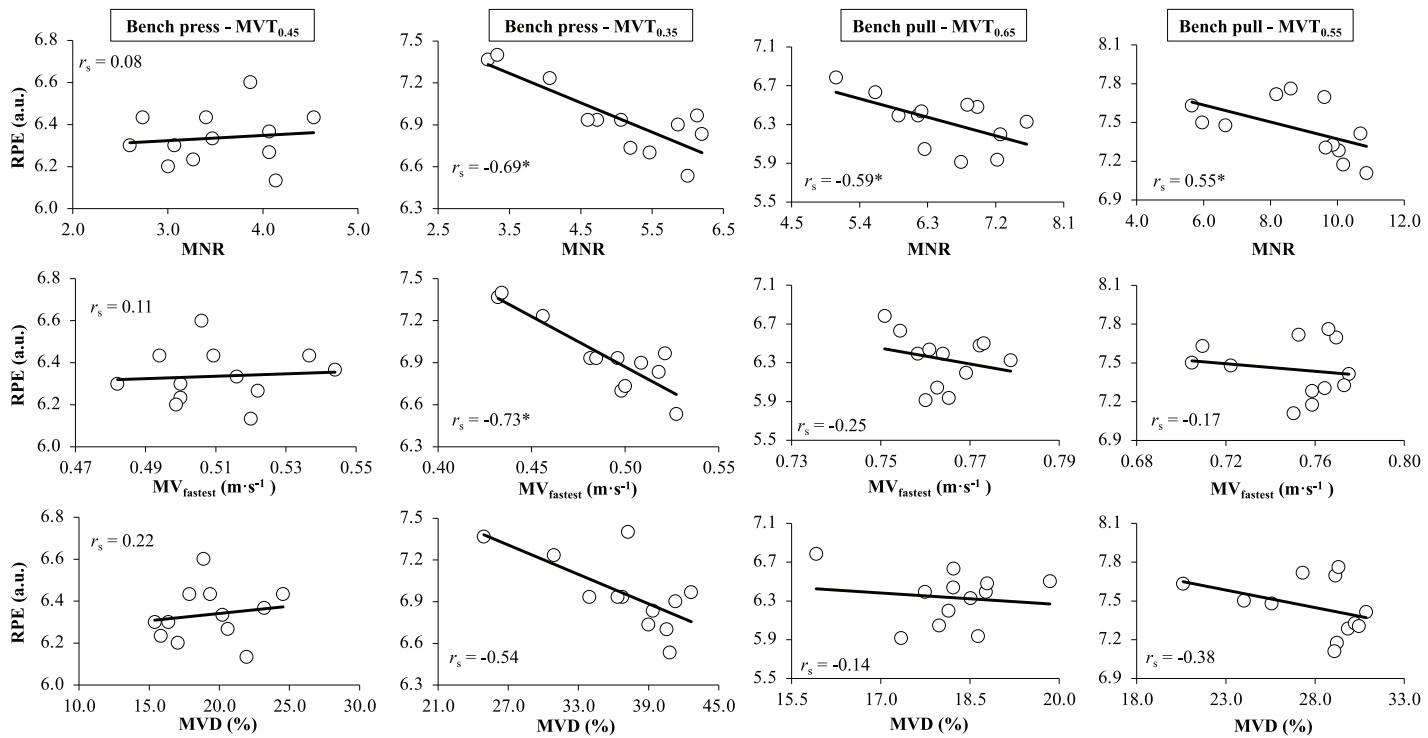

**Figure 2 Relationship of RPE with the MNR (A), MVfastest (B), and MVD (C) during the bench press and bench pull at moderate (left) and high (right) level of effort.** RPE, ratings of perceived exertion; MNR, maximum number of repetitions completed; MVfastest, fastest mean velocity of the set; MVD, mean velocity decline; MVT, minimum velocity threshold; rs, Spearman's correlation coefficient.

MNR and RPE with the only exception of the trivial $r_s$ observed during the bench press for $MVT_{0.45}$. Trivial to small correlations were commonly detected between $MV_{fastest}$ and RPE with the exception of the very large negative $r_s$ during the bench press for $MVT_{0.35}$. Finally, the associations between MVD and RPE were less consistent only observing a moderate negative $r_s$ again during the bench press using $MVT_{0.35}$.

# DISCUSSION

This study was designed to investigate the impact of inter-set rest period duration on RPE values during resistance training sets terminated upon surpassing a specific submaximal absolute MVT. The main finding was that the length of inter-set rest periods did not affect RPE values, regardless of the exercise, set number, or proximity to failure. As anticipated, increased proximities to failure, indicated by lower terminal velocities, were associated with higher RPE values in both the bench press and bench pull exercises. However, surprisingly, despite the terminal velocity being consistent across all four sets, an upward trend in RPE values was observed as the number of sets increased. Finally, while acknowledging the inconsistency in the relationship between RPE and mechanical variables (MNR, MVfastest, and MVD), it was generally observed that higher magnitudes of these mechanical variables were associated with lower RPE values, suggesting that the

perception of effort reported at the end of a set is also influenced by the fatigue experienced at the beginning of the set.

In line with our main hypothesis, RPE values were not affected by the length of inter-set rest periods. These results apparently contradict previous research that found higher RPE values with shorter inter-set rest periods (*Senna et al., 2011*; *Farah et al., 2012*). For example, *Senna et al. (2011)* compared the effects of different inter-set rest intervals (1, 3, and 5 min) on RPE values during a range of multi- and single-joint exercises, and they consistently observed higher RPE values using the shortest rest interval (1 min). One noteworthy distinction in our study is that the sets were intentionally not performed to failure but rather concluded at a predetermined submaximal level of fatigue, accomplished through the application of a specific MVT. Similarly, *Farah et al. (2012)* reported that resting for 30 s led to higher RPE values compared to resting for 90 s during a variety of exercises (bench press, knee extension, seated row, knee curl, and frontal rise) performed at 50% of 1RM. However, as the number of repetitions were matched for both rest protocols (three sets of 12, nine, and six repetitions, respectively), it is plausible that subjects approached closer to muscular failure when utilising the 30-s rest interval protocol. In contrast, our results indicate that when matching the terminal set velocity, rather than the total number of repetitions, the length of inter-set rest periods does not significantly impact the development of RPE. This study offers novel insights into the influence of inter-set rest periods on RPE during non-failure sets, further supporting the utility of absolute MVTs to elicit consistent levels of effort.

Unlike the length of inter-set rest periods, the proximity to failure and the number of sets were found to contribute to RPE development. Consistent with our second hypothesis and supporting previous research (*Robertson et al., 2003*; *Abbiss et al., 2015*; *Balsalobre-Fernandez et al., 2018*), our findings demonstrate that a closer proximity to failure (by implementing a lower MVT) produced higher RPE values during both the bench press and bench pull exercises. However, even with matched velocities for the last repetition of the set, we observed an upward trend in RPE values as the number of sets increased. These results suggest that not only is RPE affected by the terminal MVT, but also by the fatigue experienced at the beginning of the set, which is expected to be accentuated with increased number of sets. This finding is supported by previous literature which has also reported increases in RPE over multiple sets (*Pincivero et al., 1999*; *Senna et al., 2011*). However, an innovative aspect of our study is that sets were terminated upon exceeding a predetermined MVT, while in previous the number of repetitions were matched or sets were performed to failure (*Pincivero et al., 1999*; *Senna et al., 2011*). Overall, our findings suggest that the development of fatigue intra- and inter-set, rather than training density alone, contribute to RPE development as measured by OMNI-RES. This is logical considering that OMNI-RES RPE responds to metabolic stress such as blood lactate concentration (*Robertson et al., 2003*). The higher levels of metabolic stress and fatigue induced by the number of sets and proximity to failure could lead to increased firing from afferents III and IV (*Amann et al., 2011*; *Taylor et al., 2016*; *Sidhu et al., 2017*; *Alix-Fages et al., 2022*), subsequently influencing RPE values (*Broxterman et al., 2018*), and impairing movement velocity (*i.e.*, force production capabilities) across sets and repetitions due to

impaired motor neuron discharge characteristics (*Amann et al., 2011*; *Taylor et al., 2016*; *Blain et al., 2016*; *Sidhu et al., 2017*).

Since subjects terminated the sets upon surpassing a predetermined MVT while maintaining the absolute load constant across the sets, it is anticipated that their force production capacity would remain consistent at the end of each set, regardless of variations in MNR and $MV_{fastest}$. A decrease in MNR and $MV_{fastest}$ with increased number of sets is an indicator that subjects started the sets with higher levels of fatigue. As hypothesised, lower magnitudes of the mechanical variables MNR, $MV_{fastest}$, and MVD were generally associated with greater RPE values, suggesting that the perception of effort reported at the end of a set is also influenced by the fatigue experienced at the beginning of the set. Based on our findings, when matching the level of fatigue at the end of each set, subjects are expected to report greater RPE values when they initiate the set with a lower mechanical performance. Moreover, previous research has indicated that even when RPE is measured before commencing a set, higher pre-set RPE values are associated with elevated blood lactate concentrations (*Larson & Potteiger, 1997*), which can impact the overall RPE of each set. For this reason, starting a set under conditions of reduced mechanical performance or heightened metabolic stress would result in increased RPE values, even if the mechanical performance at the end of the set remains comparable. This phenomenon may be attributed to the potential increase in firing of III and IV muscle afferents, which could influence RPE starting from the very first repetition of the set. In line with our results, it was previously documented that even when using the same intensity of load, movement velocity was significantly correlated to RPE measured by OMNI-RES during the leg press exercise (*Mayo, Iglesias-Soler & Kingsley, 2019*). Similarly, *Hardee et al. (2012)* found a clear association between RPE development and the decline in power output as an expression of fatigue. Likewise, *Zhao, Nishioka & Okada (2022)* observed that RPE correspondingly changes with muscular fatigue during elbow flexion and knee extension resistance training exercises.

The present study is not free of limitations. While interpreting our findings, it is important to note that the generalizability of our results is limited to physically active adults and specifically applies to bench press and bench pull exercises. While all our study's participants were physically active and experienced in resistance training, the inclusion of both men and women in the same sample could be a potential confounding factor, albeit minimized by our crossover study design where subjects serve as their own controls. Another limitation is that the study did not consider the potential influence of other physiological markers of fatigue, such as electromyographic activity or lactate concentrations, which could provide further insights into the relationship between RPE and fatigue. However, this study presents several important strengths, including the implementation of a randomised crossover design and the integration of multiple resistance training variables encompassing two exercises (bench press and bench pull), three inter-set rest protocols (R1, R3, and R5), and two proximities to failure (farther from muscular failure and closer to muscular failure), which contribute to our understanding of RPE development in non-failure resistance training sets. However, the inclusion of

multiple conditions in our study might be seen as a limitation by some readers, as it adds complexity to the interpretation of the results.

## CONCLUSIONS

The duration of inter-set rest periods did not have a significant impact on RPE values. However, the proximity to failure and the number of sets were identified as significant factors influencing RPE development. Higher proximities to failure, indicated by lower MVTs, were associated with increased RPE values. Moreover, as the number of sets increased, an upward trend in RPE values was observed. The increase in RPE with the number of sets together with the general negative correlations observed between RPE and mechanical variables (MNR, $MV_{fastest}$, and MVD) suggest that post-set RPE values are affected by the fatigue experienced at both the beginning and end of the set. It is important to note that these results apply to non-failure sets performed with two commonly used upper-body exercises (bench press and bench pull).

## ACKNOWLEDGEMENTS

We would like to thank all subjects for their commitment to the study.

### Funding
The authors received no funding for this work.

### Competing Interests
Amador García-Ramos is an Academic Editor for PeerJ.

### Author Contributions
- Danica Janicijevic conceived and designed the experiments, analyzed the data, prepared figures and/or tables, authored or reviewed drafts of the article, and approved the final draft.
- Sergio Miras-Moreno conceived and designed the experiments, performed the experiments, authored or reviewed drafts of the article, and approved the final draft.
- Maria Dolores Morenas-Aguilar conceived and designed the experiments, performed the experiments, authored or reviewed drafts of the article, and approved the final draft.
- Pablo Jiménez-Martínez analyzed the data, authored or reviewed drafts of the article, and approved the final draft.
- Carlos Alix-Fages analyzed the data, prepared figures and/or tables, authored or reviewed drafts of the article, and approved the final draft.
- Amador García-Ramos conceived and designed the experiments, analyzed the data, authored or reviewed drafts of the article, and approved the final draft.

### Human Ethics
The following information was supplied relating to ethical approvals (*i.e.*, approving body and any reference numbers):

The Institutional Review Board of the University of Granada (2046/CEIH/2021).

## Data Availability

The raw measurements are available in the Supplemental File.

## Supplemental Information

Supplemental information for this article can be found online at http://dx.doi.org/10.7717/peerj.16754#supplemental-information.

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
