# Peer review of "Relationship between perceptual and mechanical markers of fatigue during bench press and bench pull exercises: impact of inter-set rest period length"

_PeerJ, doi:10.7717/peerj.16754_

## Round 0.1 · original submission · Major Revisions

Dear authors, please, consider the reviewer's suggestions and questions. Note that reviewer 1 sent an attached document. Regards

Reviewer 1 ·

Basic reporting

Authors present an interesting and well-designed study examining the influence of the duration of the between sets rests, the number of sets and the MVTs on RPE values.

Experimental design

The design is clear and correct. Nevertheless, some aspects should be clarified (see the comments attached)

Validity of the findings

No doubt these results are very practical for practitioners around resistance training

Annotated reviews are not available for download in order to protect the identity of reviewers who chose to remain anonymous.

Reviewer 2 ·

Basic reporting

The paper “Relationship between perceptual and mechanical markers of fatigue during bench press and bench pull exercises: impact of inter-set rest period length” aimed to explore whether the relationship between perceptual and mechanical variables is affected by the length of inter-set rest periods during resistance training sets not leading to failure. The article is interesting; however, the authors adopt a model with manipulation of different variables, which generates complexity and biases in the interpretation of the primary objective of the study (“to explore whether the relationship between perceptual and mechanical variables is affected by the length of inter-set rest periods during resistance training sets not leading to failure”).

Experimental design

1. Line 131: what was the maximum?
2. Line 147: “regularly engaged in physical activity”… But were they performing resistance training regularly? How long had they been strength training regularly?
3. The protocol of 12 sessions with at least 48 hours between sessions comprises a minimum period of 24 days. Did the authors perform any control over the effect on women's menstrual cycles?
4. Did the authors apply any pre-session test/control? Any perceptual or physiological measures to identify the state of recovery/readiness before the sessions?
5. Was there any control/standardization of nutritional intake during the 12 sessions?
6. Did the authors consider applying a two-way ANOVA (inter-set rest protocol and level of effort) and applying the necessary corrections?

Validity of the findings

7. I would like to hear from the authors before proceeding with the evaluation.

Additional comments

8. Line 94: add a “.” after “et al”
9. Line 109: delete “(1, 3, or 5 minutes)”
10. Line 110: delete “(MVlast = 0.45 and 0.65 m.s-1, respectively)”
11. Line 111: delete “(MVlast = 0.35 and 0.55 m.s-1, respectively)”
12. Line 116: consider to delete “This hypothesis is justified because MVlast, which represents force production capabilities at the end of the set, was the same for the three inter-set rest protocol”.
13. Overall, the introduction is well written. However, it is a little extensive. Consider changing to a more to-the-point introduction. The details of some studies cited can be presented in the discussion.

---

## Round 0.2 · Minor Revisions

Dear authors,

Thank you for adressing the issues raised. The reviewers provided opposing recommendations and, I believe, both have relevant points regarding your work. That said, I'm not convinced that a "fatal flaw" is present but, before a final acceptance, I would like to request some remaining issues. Thus, please consider the following suggestions:

L52. Please, add the missing quotation mark.
Procedures: Please add the rationale for choosing these specific MVT for each exercise value, with a proper reference.
L174. For reader's clarity, please, inform explicitly that 1RM tests were repeated at the beginning of all 12 sessions.
Results: The authors included the MVT actual values (L183), please also provide the MNR and MVfast information.
Discussion: One of the reviewers recommended rejection based on sample heterogeneity and the complexity of multiple variables involved. Please, consider including a statement in the limitations paragraph regarding these points.
Figures: Please, be consistent with the axis values on the graphs. Also, please review the use of commas vs periods in your axis values.

Reviewer 1 ·

Basic reporting

Authors have satisfactorily addressed all the comments and requirements I requested. Therefore, I believe that the manuscript is now ready for acceptance. Congratulations to the authors!

Experimental design

The good job done by the authors throughout the review process has strengthened the information regarding the experimental design.

Validity of the findings

The good job done by the authors throughout the review process has strengthened the information regarding validity of the findings.

Reviewer 2 ·

Basic reporting

The study is ell written.

Experimental design

I recognize that the revised version is better balanced and well written. However, I am not convinced that the manipulation of different variables, as well as the use of a heterogeneous sample, cannot generate confounding biases in the interpretation of the results.

Validity of the findings

I recognize that the revised version is better balanced and well written. However, I am not convinced that the manipulation of different variables, as well as the use of a heterogeneous sample, cannot generate confounding biases in the interpretation of the results.

---

## Round 0.3 · accepted · Accept

Dear authors,

Thank you for addressing the issues raised by the reviewers. I'm happy with the process and the current version. Congratulations.